# Long-Term Variations in Spring Water Mean Transit Time in a Forested Headwater Catchment in Japan

**Isabela Silveira Baptista** [1,*] **, Maki Tsujimura** [2] **and Yuichi Onda** [2]

1    Graduate School of Science and Technology, University of Tsukuba, Tsukuba City 305-8577, Japan
2    Faculty of Life and Environmental Sciences, University of Tsukuba, Tsukuba City 305-8577, Japan
*    Correspondence: s2030291@s.tsukuba.ac.jp

**Abstract:** In this study, we applied a new methodology that utilized stable isotopic concentrations of water and $SF_6$ concentrations of the air to investigate temporal variations in the spring water mean transit time (MTT) of a forested headwater catchment in Japan. We monitored the area from June 2010 to November 2021, including the forest thinning that occurred in October 2011. The $SF_6$ concentrations were used to estimate the apparent age of the spring water, which was used to initiate a parameter calibration that determined the best possible MTT for each study period. The MTT was evaluated by estimating the d-excess variations of the spring water stable isotopes using the exponential piston flow model. The MTT ranged from 40 to 55 months and tended to increase soon after the forest was thinned. In contrast, the MTT was shorter by approximately 6 years after thinning occurred. The results indicate that the characteristics of the hydrological processes in the catchment underwent long-term changes after the forest was thinned. This study demonstrates that using combined tracer methods to investigate the hydrological response to forest management practices improved the results and can be used for better forest and subsurface water resource management.

**Keywords:** mean transit time; spring water; stable isotopes; $SF_6$; tracers; forest thinning





## 1. Introduction

The Japanese archipelago is covered by forests that are mainly located in mountainous areas, wherein forested headwaters play a substantial role in supplying water to lowland areas [1]. Almost half of these forested areas are plantation forests, mainly of Japanese cedar and Japanese cypress, and have a lack of management, which can result in a wide canopy coverage that increases soil erosion, for instance [2]. According to [3], forest restoration and soil restoration dynamics, as well as hydrological process conservation, are sparsely studied. Additionally, [4] stated that water scarcity or abundance is an important factor in forest degradation, and no detailed analysis of forest degradation and soil characteristics has been conducted.

To address these management deficiencies and improve water conservation, forest thinning could be applied [5]. Forest thinning can alter evapotranspiration and runoff in catchments, which changes the local water balance and flow paths, and affects water availability, quality, and supply [2,6].

To properly understand how forest thinning affects hydrological processes, it is important to know the mean transit time (MTT) of spring water over a long period. Transit time is the amount of time elapsed from when water enters a system until it leaves the system, e.g., from when precipitation infiltrates into a catchment until it is discharged as spring water [7]. The MTT can be modeled by the relationship between precipitation and hydrological components using conservative tracers [8–10].

Previous studies have shown that estimating the MTT using multi-tracer analysis provides complementary insights into hydrological processes [10–12]. In particular, sulfur hexafluoride ($SF_6$) concentrations in the air and water stable isotopes ($^2H$ and $^{18}O$) are

commonly used tracers, and have been applied effectively in multi-tracer studies [13,14]. However, the combined application of $SF_6$ and $^2H/^{18}O$ in forests undergoing thinning has not been addressed previously.

The chemical stability of $SF_6$ and its continuously increasing concentrations in the Northern Hemisphere make it an effective tracer for groundwater recharge [15–17]. Therefore, a multi-tracer approach could improve the robustness of age interpretations and the current understanding of groundwater recharge processes.

The main objective of this study was to clarify how the characteristics of a forest alter its short and long-term hydrological processes in a headwater catchment. In this study, we argue that estimated MTT provides useful information regarding the effects of forest thinning, as MTT offers vital details on hydrological processes, including storage at a catchment scale [7] and initial approximations of catchment transport behavior [18]. We applied a new multi-tracer methodology that included analyses of stable isotopes of water and $SF_6$ concentrations of the air. Previous studies have mainly used trial-and-error approaches to identify the water age [7–10]. This study aims to provide more reliable MTT estimates, as the model parameters are calibrated beforehand using apparent age information based on the $SF_6$ data. Using past and current hydrological and tracer data, we sought to effectively understand the short- and long-term temporal changes in MTT after forest thinning in Japan.

## 2. Materials and Methods

### 2.1. Study Area

This study was conducted in a small forested headwater catchment composed primarily of plantations in the Mount Karasawa Experimental Forest (Tokyo University of Agriculture and Technology) in Tochigi Prefecture, Japan (Figure 1). The plantations comprise artificial forests of Japanese cedar (Sugi) and Japanese cypress (Hinoki) trees that were planted during the 1960s.

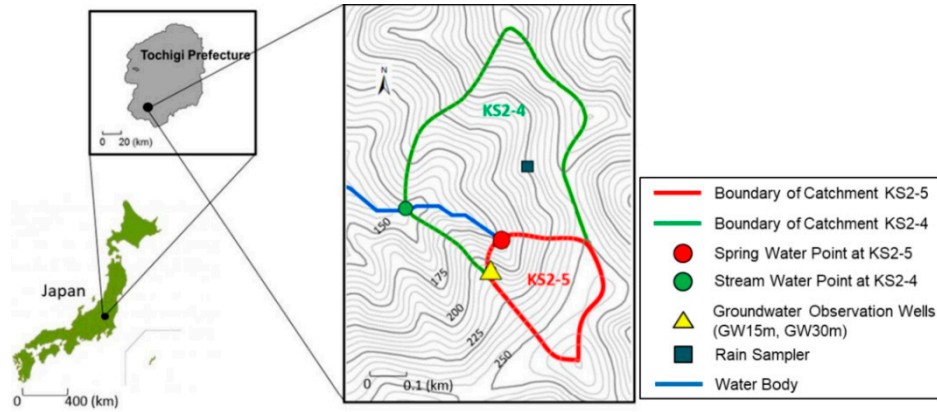

**Figure 1.** Map of the study area showing the KS2-5 catchment (red boundary) where the spring water and groundwater data were collected, as well as the nearby KS2-4 catchment (green boundary) where precipitation data were collected.

The spring water collection site was at the KS2-5 (1 ha) catchment, near the boundary to the K2-4 (5.1 ha) catchment, and had a mean slope gradient of 55%. Rainwater was collected at the nearby K2-4 catchment. Two observation wells were installed in the KS2-5 catchment, with depths of 15 m and 30 m, respectively (GW 15 m and GW 30 m, respectively). The soil in the study area is brown forest soil with a layer thickness ranging from 0.35 m to 2.0 m. The saturated hydraulic conductivity of the soil layer at 0.5 m ranges from $2.24 \times 10^{-6}$ to $2.30 \times 10^{-6}$ m/s. The bedrock underlying KS2-5 catchment consists alternately of strongly weathered sandstone and weathered sandstone, which is classified as part of the Chichibu Paleozoic Formation.

The climate in this area is humid temperate, with a rainy season that extends from late May to July, and the typhoon season from late August to October [16]. The mean annual precipitation and air temperature during the past two decades were approximately 1239 mm and 14 °C, respectively [18].

### 2.2. Study Period

Forest thinning management throughout the Mount Karasawa area included tree cutting in July 2011 and landing and transporting in October 2011 [18]. In the study area, 50% intensity linear thinning was performed in the KS2-5 catchment forests in October 2011. The study period was divided into three parts: "before thinning" (June 2010–September 2011), "soon after thinning" (November 2011–December 2013), and "long after thinning" (August 2017–November 2021).

As mentioned above, the aim of this study was to understand the short- and long-term temporal variations in MTT to shed light on the hydrological changes associated with pre- and post-tree thinning treatments.

### 2.3. Sampling and Data Analysis

Continuous monthly hydrological observations and sampling were conducted during each field survey. Spring water, rainwater, and groundwater (from GW 15 m and GW 30 m) were sampled for stable isotope and $SF_6$ analyses.

The water samples were collected in 100 mL plastic containers (stable isotope analyses) and 500 mL amber glass bottles ($SF_6$ analyses). To collect the water sample for $SF_6$ analysis, the bottle and cap were placed in a 2 L volume bucket with water overflowing from it. When sampling, it was necessary to prevent contamination with the atmosphere, so we pumped up the water at the spring point and groundwater well and confirmed that there was no air intrusion in the samples.

Groundwater levels were observed at intervals of 10 min to 1 h. A water level/temperature logger (HOBO CO-U20-001-01, Tempcon Instrumentation Ltd., Ford, UK) was used for the observations and another one of the same type was used to correct the atmospheric pressure from the measured water, which was installed near the observation wells.

The flow rate of spring water was measured directly below the water sampling point. Water was sampled with a 45 L plastic bag for 1 to 5 min, and the flow rate was measured with a graduated cylinder. By repeating the measurement three times or more, the average flow rate was calculated.

A 100 mL glass syringe (glass syringe, VAN) and a three-way stopcock (Telfation three-way stopcock L-lock connector (TS-TL2K), TERUMO) were used to collect air samples in the catchment area. After sealing with a three-way stopcock, the tube was kept pressurized with vinyl tape and brought back to the laboratory.

Both the stable isotopes and $SF_6$ dissolved concentration analyses were done in the Hydrology and Water Environment Laboratory, Faculty of Life and Environmental Sciences, University of Tsukuba, Japan.

Because air masses over Japan cause seasonal variations in rainfall, we used the d-excess of $^2H$ and $^{18}O$ (stable isotopes of water) as conservative tracers to estimate the MTT [6]. Stable isotope ratios were analyzed using a liquid water isotope analyzer (L2120-i, Picarro) using wavelength scanning cavity ring-down spectroscopy. The results are expressed as $\delta$ values (‰) as shown in Equation (1) [19]:

$$\delta\ (‰) = \frac{R_{Sample} - R_{SMOW}}{R_{SMOW}} \times 1000 \tag{1}$$

$R_{SMOW}$ indicates $^{18}O/^{16}O$ or $^2H/^1H$ in VSMOW (Vienna Standard Mean Ocean Water), and $R_{Sample}$ indicates $^{18}O/^{16}O$ in the sample. The analytical accuracy is $\pm 0.3‰$ for $\delta^{18}O$ and $\pm 1.0‰$ for $\delta^2H$.

The analysis system of $SF_6$ consists of a separation/concentration device that separates $SF_6$ gas from the water sample and then concentrates it, and a data processing device. In this operation, the limit of quantification in $SF_6$ analysis is 0.1 fmol. The same analytical

system was used for the analysis of dissolved concentrations of SF$_6$ in air. Spring water and groundwater samples were analyzed 2 to 4 times, and air samples 4 to 5 times. The analytical accuracy of the gas in air and water samples is less than 1.5% and 3%, respectively [17].

### 2.4. Apparent Ages

We estimated the apparent ages of the spring water samples using dissolved SF$_6$ concentrations in spring water that were converted to atmospheric mixing ratios based on Henry's law, according to the International Atomic Energy Agency [20] guidelines, and the purge-and-trap gas chromatography procedure described by Busenberg and Plummer [15].

This dating method assumes that the SF$_6$ concentration in the water sample at the time of recharge is preserved. This value is used to correct the atmospheric conversion. Using the average SF$_6$ concentration of the air in the study area during the observation period, the northern hemisphere data were proportionally distributed and corrected, and a correction line (Mt. Karasawa line) was generated. The atmospheric SF$_6$ concentrations measured in the study area differed by approximately 3–5 pptv from northern hemisphere values (Figure 2).

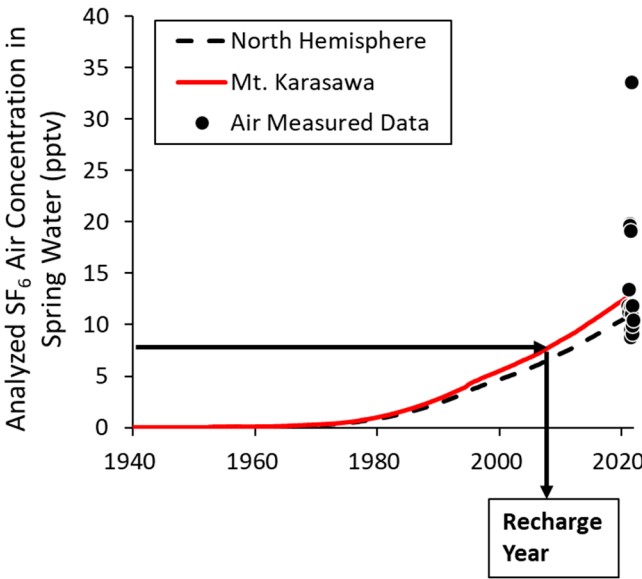

**Figure 2.** Air conversion lines (Mt. Karasawa area and North Hemisphere), according to the SF$_6$ concentrations in the air obtained in the study area, which indicate the recharge year of each water sample.

The apparent ages of the water samples can be estimated by comparing them with the atmospheric SF$_6$ concentrations (indicated by the air line in Figure 2). Using the concurrent spring water samples and SF$_6$ concentrations in the air, we determined when that sample of spring water was recharged, from which we calculated the apparent age of each sample.

We used the SF$_6$ apparent age data to have an initial estimation of the spring water age. This initial estimation is considered as the first parameter in the model calibration and fitting process that is part of the parameter search shown in the following subsections.

### 2.5. Mean Transit Time (MTT) Estimation Model

The d-excess of water samples expresses the close correlation between $^{18}$O and $^{2}$H in precipitation which can be demonstrated by the global meteoric water line, which has the general form as follows [9]:

$$\delta 2H = 8 \times \delta 18O + d \tag{2}$$

where the offset $d$ is called deuterium excess or d-excess. In Japan, the air masses that cause rainfall have seasonal variations, which make them easier to target and analyze; in this study, the *d*-excess value is used as a tracer.

Maloszewski and Zuber [21] showed that for a system with a steady flow, the output isotopic tracer is related to the input isotopic tracer by the following convolution integral:

$$\delta out(t) = \int_0^\infty g(t')\delta in(t - t')\mathrm{d}t' \tag{3}$$

where $\delta out(t)$ is the output signature, $t'$ is the entry time into the system, $t$ is the calendar time, $\delta in$ is the input signature, and $g(t')$ is the transit time distribution. The $g(t')$ used in this study was the exponential piston flow model (EPM). This model was selected after comparing different models to best explain the characteristics of the study area. The EPM exhibited the smallest error for each parameter and could be used to describe the study catchment.

According to findings in [9], the EPM can be expressed as:

$$g(t) = \frac{\beta}{\tau} \exp\left(-\frac{\beta t}{\tau} + \beta - 1\right) \tag{4}$$

where $\beta$ is the total water volume divided by the volume with an exponential distribution of transit times, $t$ is the calendar time, and $\tau$ is the MTT.

We determined the smallest possible root mean square error (RMSE) of the model to estimate the similarity of the calculated output to the observed data, as follows:

$$\mathrm{RMSE} = \sqrt{\frac{\sum_{i=1}^{n}(Oi - Xi)^2}{n}} \tag{5}$$

where $Oi$ is the $i$th observed value, $Xi$ is the corresponding value calculated by the model, and $n$ is the sample size.

The water transit time is equivalent to the total storage divided by the water flux [3]; therefore, we obtained the storage volume using:

$$\mathrm{MTT} = \mathrm{Storage\ Volume}/\mathrm{Flow\ Rate}, \tag{6}$$

where the storage volume (mm) is the amount of water stored in the catchment during each year of the study period, and the average discharge (mm/d) is the amount of runoff from the spring site, which was assumed to be an average value per year. The MTT values determined in this study are expressed in months.

### 2.6. Model Calibration and Fitting Process

The spring water MTT was estimated using the $SF_6$ apparent age-weighted average to calibrate the EPM parameters according to the temporal variations in the d-excess of the stable isotopes.

To find the pair of parameters ($\beta$, $\tau$) we follow the subsequent two-step method:

- The first step is to find $\beta$. We search in the range of 1 to 2, in increments of 0.01. We consider the initial $\tau$ equal to the $SF_6$ apparent age weighted average (From the $SF_6$ apparent age primary estimation). We select the $\beta$ based on the value with the lowest RMSE
- Then, the second step is to find $\tau$. We search in the range of 1 to 100, in increments of 0.1. We select $\tau$ that is associated with the lowest RMSE.

We used R Studio, which is a software for statistical computing, to optimize the parameters search. This way, the data can be calculated automatically following the parameter search range as shown above and the best fitted ones are found. Thus, the pair of parameters ($\beta$, $\tau$) is the one following this two-step methodology.

The MTT was determined for each study period in which experimental investigations were conducted in the study area. This study has an initial parameter that can be found from previous data analysis and, therefore, it is an initial estimation based on theoretical

knowledge, which is therefore likely to be correct. Since previous studies have exclusively used trial-and-error approaches, we propose that this new method provides more reliable MTT estimates.

## 3. Results

### 3.1. $SF_6$ Apparent Age Weighted Average

As we have seen on Figure 2 previously, the $SF_6$ concentrations of 17 spring water samples collected from 2020 to 2021 were investigated. After analyzing the measured air concentrations of $SF_6$ in the Mount Karasawa area and those in the spring water samples, we constructed the lines shown on the figure. Since we had the $SF_6$ air concentrations for the corresponding spring water samples, we could obtain each sample's recharge year by comparing it to the Mount Karasawa line. Therefore, we obtained the apparent age of each sample, and the average value weighted by the discharge of the sampling day.

Table 1 shows the apparent age of each sample and the spring water daily discharge of their sampling days. The samples (apparent age) are weighted to better match the population "weights" (daily discharge), so every term has a corresponding weight. The weighted average considers the correlation of spring water's apparent age and its discharge. Hence, each sample of spring water from a specific field survey day is associated to the daily discharge of spring water on that same day. This average of the apparent age is an initial estimation based on theoretical knowledge, and is part of the parameter search and fitting process. Finally, there is one $SF_6$ apparent age that can be considered as the initial $\tau$ of the model parameter calibration and fitting process.

**Table 1.** Sampling dates, apparent $SF_6$ age of each sample (in months), and their daily discharges (in mm/day). The weighted average of the apparent ages is 39.1 months.

| Date (y/m/d) | $SF_6$ Apparent Age (Months) | Discharge (mm/Day) |
|:---:|:---:|:---:|
| 2020/11/26 | 252 | 0.12 |
| 2021/2/17 | 204 | 0.10 |
| 2021/3/31 | 126 | 0.11 |
| 2021/4/14 | 66 | 0.09 |
| 2021/5/12 | 42 | 0.05 |
| 2021/5/26 | 78 | 0.06 |
| 2021/6/9 | 78 | 1.02 |
| 2021/6/23 | 60 | 0.74 |
| 2021/7/7 | 24 | 0.97 |
| 2021/7/21 | 36 | 0.11 |
| 2021/8/4 | 66 | 0.14 |
| 2021/8/19 | 6 | 2.55 |
| 2021/9/28 | 66 | 0.27 |
| 2021/10/13 | 30 | 0.34 |
| 2021/10/28 | 6 | 0.22 |
| 2021/11/10 | 6 | 0.30 |
| 2021/11/24 | 6 | 0.09 |

Considering the spring water discharge, the weighted average of the apparent age of the samples was 39.1 months, which was the initial estimation used for parameter calibration ($\tau$). As described later, this result was close to the MTT from the "before thinning" and "long after thinning" periods.

### 3.2. Hydrological Data

Figure 3 shows the long-term fluctuations in precipitation and runoff of the catchment stream water from April 2010 to November 2021. These fluctuations did not include substantial changes in the annual rainfall curve before or after thinning was performed. Forest thinning occurred in October 2011. The stream water runoff was relatively high in June and July (70–100 mm/d) when rainfall was also high, and was relatively low in

January and February (0.001–0.5 mm/d) when precipitation decreased. This indicates that seasonal steady-state fluctuations occurred annually.

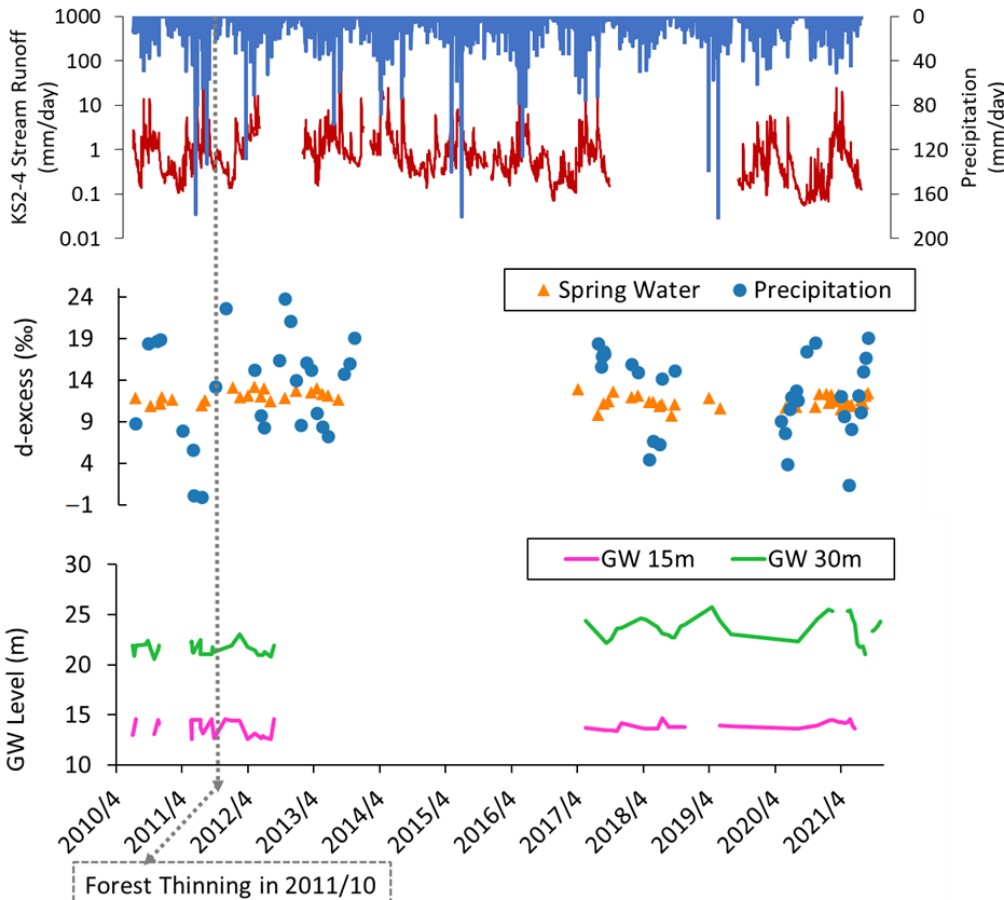

**Figure 3.** Stream water precipitation and runoff measurements, groundwater (GW) levels measured in the two boreholes, and spring water d-excess (‰) during the observation period (June 2010–November 2021). The forest was 50% thinned in October 2011.

Figure 3 also shows the spring water and precipitation d-excess values from June 2010 to November 2021. The fluctuations in the precipitation d-excess were large, ranging from −1‰ to 23‰ owing to the seasonal rainfall variations in the study area. In contrast, the spring water d-excess fluctuated less widely (9–13‰).

In addition, the groundwater level in the GW 15 m observation well was steady over time, with a slight increase soon after thinning (Figure 3). The groundwater level in the GW 30 m observation well exhibited wider oscillations throughout the years, including a large decrease after 2019. The GW 30 m groundwater level increased considerably under higher precipitation, but the response speed to rainfall was delayed by about 2–3 days compared to the GW 15 m observation well. The fluctuations in groundwater level were large (about 2 m); however, the response speeds of the observation wells to rainfall differed. When compared to the stream runoff, we found that the increasing flow rate exhibited a similar tendency as the increases in groundwater level.

The water balance at the Mount Karasawa study area can be understood through Table 2, which shows the annual amounts of precipitation (P), stream discharge (Q), and water loss (L) in mm per year, from 2010 to 2021. The water loss is taking into consideration the analysis from [6], in which the estimated L = P − Q. The years 2018 and 2019 have no discharge data due to strong typhoon events in those years that have made the data unable to be calculated properly.

**Table 2.** Annual water balance at the Mount Karasawa study area.

| Year | Precipitation (mm) | Discharge (mm) | Water Loss (mm) |
|------|--------------------|-----------------|------------------|
| 2010 | 1197.9 | 241.9 | 955.9 |
| 2011 | 1411.0 | 726.6 | 684.4 |
| 2012 | 1295.2 | 495.6 | 799.6 |
| 2013 | 1207.8 | 759.5 | 448.3 |
| 2014 | 1419.8 | 923.4 | 496.3 |
| 2015 | 1342.2 | 843.3 | 498.9 |
| 2016 | 1434.3 | 390.6 | 1043.7 |
| 2017 | 1322.8 | 341.2 | 981.6 |
| 2018 | 979.9 | - | - |
| 2019 | 1291.2 | - | - |
| 2020 | 1092.2 | 261.6 | 830.6 |
| 2021 | 1189.6 | 328.7 | 860.9 |

As we can see in the Table 2, the precipitation trend is stable throughout the years. The stream water discharge shows an increasing trend in the years soon after thinning. In the years long after thinning, especially from 2016, the stream water discharge shows a decreasing trend.

The water loss, in which evapotranspiration is a part of it, shows a decreasing trend on the years soon after thinning. On the years long after thinning, especially from 2016, the water loss shows an increasing trend.

### 3.3. Spring Water Mean Transit Time

We estimated the spring water MTT using the EPM, as shown in Figure 4. The d-excess of the observed precipitation data, which was included in the model as the input content, and its fitted sinusoidal line is also shown in the figure. The d-excess of the spring water was plotted from June 2010 until November 2021. After calibrating the model parameters and obtaining the smallest RMSE, we fit the calculated d-excess values to the observed data. The d-excess calculated using the model and the observed data exhibited long-term sinusoidal variations (Figure 4). The monthly spring water d-excess values were obtained from the stable isotope data at each sampling point. The final EPM parameters are also shown in Figure 4, under the graph representing each study period, "before thinning", "soon after thinning", and "long after thinning".

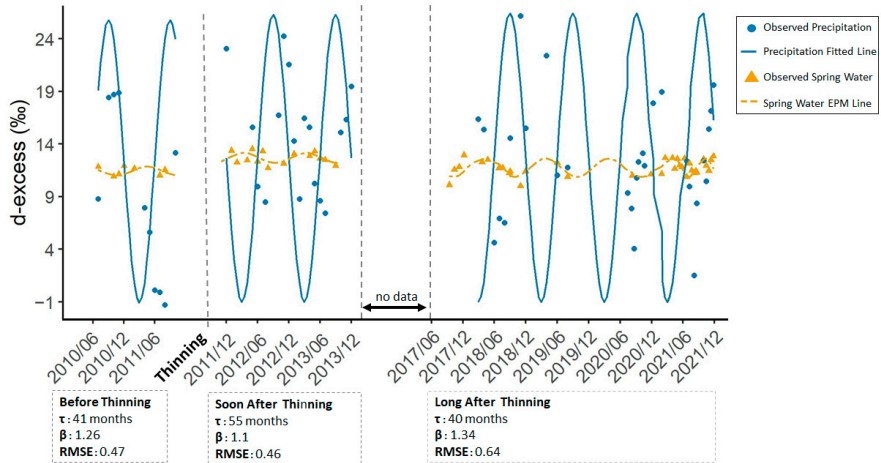

**Figure 4.** Long-term variations in the spring water d-excess and the mean transit time (MTT) analysis flow determined by applying the EPM to fit the observed data. The final $\tau$ and $\beta$ parameters and the root mean square error (RMSE) of each study period are shown in their corresponding boxes under the graph.

According to the $\tau$ results from the model (Figure 4), the MTT increased after forest thinning was performed. Six years later, the MTT decreased again. We understand that with a higher infiltration soon after thinning, older groundwater mixed with the younger and was pushed out of the underground system. The MTT during the final study period was close to the $SF_6$ apparent age. The lowest RMSE was 0.45 in the "soon after thinning" period.

*3.4. Long-Term Hydrological Processes in the Study Area*

Based on the MTT and the average annual spring water discharge, we determined the storage volume of the catchment over time (Equation (6)). As shown in Figure 5, the average volume of the catchment increased from 676 mm to 1171 mm 1–2 years after the forest was thinned. Approximately six years later, the storage volume was decreased to 588 mm.

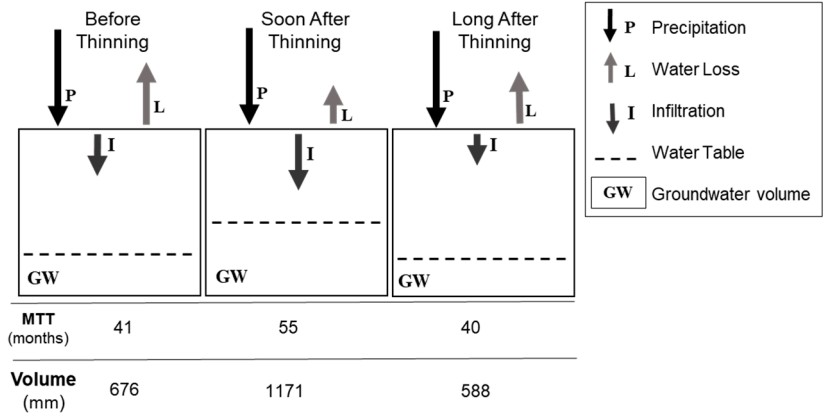

**Figure 5.** Schematic diagram illustrating hydrological processes in the study area during the "before thinning" (2010–2011), "soon after thinning" (2011–2013), and "long after thinning" (2017–2021) study periods. Arrow lengths demonstrate changes in the amount of water loss and infiltration, with the precipitation arrow remaining stable. The water table line indicates fluctuations in groundwater storage. The MTT and volume are shown below the diagram.

In the proposed schematic diagram of the study area, considering stable precipitation over time and the water balance analysis of the study area, changes in groundwater storage are explained by differences in the MTT and fluctuations in water loss and infiltration. Soon after the forest was thinned, infiltration and groundwater storage increased, while water loss decreased. During the years long after thinning, the opposite trend was observed.

## 4. Discussion

This study aimed to determine how the characteristics of a forest alter short- and long-term hydrological processes in a forested headwater catchment, considering a new multi-tracer methodology. Previous studies [10,12,22], even when considering different tracers and methods, analyzed the water's MTT or mean residence time during a short period of time (1 to 2 years), which meant that there was still a need to understand the long-term changes after forest change.

Although previous studies have shown that water transmitted by the canopy as throughfall or stemflow can have different isotopic composition than rainfall [23], throughfall isotopic shift is negligible in our study area. MTTs of groundwater and streamflow have been effectively estimated by others using rainfall isotopic signals [10]. In addition, our rainfall data set, longer than that of throughfall, was more appropriate for estimating short- and long-term MTT.

According to the findings in [17], the $SF_6$ ages of springs undergo seasonal changes, which we also observed in this study. The $SF_6$ water age analyses in [15,24–26] were similar to those conducted in this study, as they were using a short sampling period to analyze and obtain $SF_6$ ages, mean ages obtained found from a mixed-age sample, and mean ages that

were flow-weighted from the discharging water. In addition, large annual variations in the $SF_6$ age have been observed.

However, differently from previous studies, we have applied this new methodology according to the $SF_6$ apparent age data as it is an indicator of the real water age of the samples. This way, the parameters search, model calibration, and fitting process are not calculated exclusively by trial-and-error, as there is an initial parameter that can be found from previous data analysis and, therefore, it is an initial estimation based on theoretical knowledge. That is why we consider the $SF_6$ apparent age weighted average as an initial facilitator of the modeling process, but not as the final result itself.

The apparent age and MTT are a result of mixture of multiple flow paths with a different apparent age/MTT, so they tend to change temporally even during the time scale of rainstorm events. Additionally, MTT was estimated using multiple groundwater flow paths that have different transit times and could change daily, seasonally and annually depending on the mixing ratio of the different groundwater flow paths. Considering that we wanted to understand the long-term temporal variations in MTT, there was a need to properly organize the data to observe any changes that occurred during the entire study period.

Therefore, we set our research question as to verify and understand the short- and long-term temporal changes in MTT after the forest had a thinning treatment, thus we believe that the focus of the results should be on three periods: "before thinning" (June 2010–September 2011), "soon after thinning" (November 2011–December 2013), and "long after thinning" (August 2017–November 2021).

Although some studies have shown that regional water budgets may be positively affected by increasing forest and wetland cover [27], we understand that this field of study depends on the area in which it is being investigated and its specific characteristics. Different hydrological responses to forest disturbances in distinct watersheds are also due to their particular characteristics [28].

The annual water balance information supports the results of this study, which are similar to other studies in small headwater forested catchments and even global ones [29,30], which show higher runoff and groundwater levels soon after thinning of forests [6].

Our study shows that an increasing MTT reflects the increased infiltration, groundwater levels, storage volume and the decreased water loss. Since we could estimate a decrease of MTT long after thinning, this indicates a change of volume storage declining six years after thinning, which is an opposite tendency from the previous study period.

Thus, we understand that "long after thinning" the hydrological processes return to conditions similar to "before thinning". A previous study of a similar condition area showed that the effects of forest thinning on evapotranspiration decreased months after this treatment was performed [6]. As they mention in their study, it is possible this is due to interception and transpiration recovering from regrowth of understory vegetation. They concluded that it is not clear if thinning will result in net water gain when both canopy and understory vegetation will regrow rapidly after thinning.

As has been shown in our study, these results [6] and ours can confirm that the "long after thinning" period has lower effects from the forest thinning treatment, such as reduced runoff, groundwater level, and MTT, making it comparable to the "before thinning" period. Changes in vegetation cover could also have occurred in the catchment throughout the study period. Such variations may be associated with an understory vegetation post-recovery phase, as forest thinning had occurred several years prior.

Therefore, we suggest that future studies should include information on vegetation cover recovery in addition to hydrological data. Consequently, we stress that the effects of forest thinning on hydrological processes in small headwater catchments are extensive. The impacts caused by thinning were observed soon after thinning occurred because of the MTT, followed by increases in storage volume during the following two years. In contrast, six years after thinning, the impacts were smaller and hydrological processes returned to their original status trend, as demonstrated by the lower MTT and storage volume values.

## 5. Conclusions

The purpose of this study was to apply a new methodology to determine how the characteristics of a forest alter short- and long-term hydrological processes in a forested headwater catchment. Previous studies have mainly investigated changes in hydrological processes soon after forest thinning was performed. We used a combination of water stable isotopes and $SF_6$ concentrations in the air and water to investigate temporal variations in the spring water MTT and the effects of forest thinning on short- and long-term hydrological processes.

The results indicate that the MTT was 41 months before thinning. Soon after thinning, the MTT increased to 55 months. This indicates that a reduction in forest interception occurred, leading to increased infiltration, groundwater levels, storage volume and mixing with older groundwater. In addition, the MTT decreased to 40 months long after thinning, which indicates the opposite trend of decreasing infiltration, groundwater levels, and storage volume and mixing with older groundwater.

This study demonstrates that a multi-tracer approach can assess and enhance the results of short- and long-term variations in MTT and major hydrological processes. Moreover, we found that the characteristics of the hydrological processes underwent extended periods of changes following forest thinning. This knowledge is fundamental, as this topic has not been studied sufficiently. Investigations of hydrological responses to forest treatments demonstrate how the MTT and groundwater storage volume are quantitatively altered, which is essential for managing subsurface water resources in forested regions.

**Author Contributions:** Conceptualization, Funding acquisition, Writing—review and editing: M.T.; Conceptualization, Formal analysis, Investigation, Writing—original draft preparation: I.S.B.; Funding acquisition, Project Administration: Y.O. All authors have read and agreed to the published version of the manuscript.

**Funding:** This study was supported by the Japan Science and Technology (JST), Core Research for Evolutional Science and Technology (CREST) projects entitled "Field and modeling studies on the effect of forest devastation on flooding and environmental issues" and "Development of Innovative Technologies for Increasing in Watershed Runoff and Improving River Environment by Management Practice of Devastated Forest Plantation". Part of this study was supported by the Japan Society for Promotion of Science (Grant No. 16H02556 and No. 16H03111).

**Data Availability Statement:** Not applicable.

**Acknowledgments:** The authors would like to thank the Ministry of Education, Culture, Sports, Science, and Technology (MEXT) of Japan for their scholarship support during the doctoral program of the corresponding author. The authors are grateful for the infrastructure at Mount Karasawa Experimental Forest, Tokyo University of Agriculture and Technology, and the facilities at the Hydrology and Water Environment Laboratory, Faculty of Life and Environmental Sciences of the University of Tsukuba, Japan. Finally, we would like to thank Takashi Gomi and Chen-Wei Chiu, from the Tokyo University of Agriculture and Technology, for their support in this study.

**Conflicts of Interest:** The authors declare no conflict of interest.

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
