# Peer review of "Long-Term Variations in Spring Water Mean Transit Time in a Forested Headwater Catchment in Japan"

_water, doi:10.3390/w14233925_

Round 1

Reviewer 1 Report

I found this manuscript well written and well structured. The study method is also appropriate. However, there are a few things that need to be fixed, which are annotated on the attached file.

Author Response

Thank you for giving us the opportunity to revise and submit our manuscript  for publication for the Water journal. We have revised the manuscript according to the reviewers’ comments and would like to thank you for your support. Our point-by-point responses to the reviewers’ comments are in the attachment.

Reviewer 2 Report

It is an interesting study, certain points need to be improved.

Specific comments:

L26-27: Where does the claim that lack of management affects the hydrological cycle of forests come from?

L26-29: Are the authors referring to natural forests or plantations? The authors state that tree thinning could help improve current management deficiencies, but this in turn would affect water availability, quality and supply. Indirectly I understand what the authors want to convey but it is not worded correctly.

L39-40: This sentence is unnecessary; you can delete it.

L42: Perhaps “provide complementary insights” reads better

L43: Change “2H/18O” by “water stable isotopes (2H and 18O)

L66: This line indicates that spring water was sampled at K2-4, but the map in Figure 1 depicts stream water.

L69-70: Could you describe some physical properties of the soil in the study area.

L86-90: This paragraph can be changed to: “As mentioned above, the aim of this study was to understand the short- and long-term temporal variations in MTT to shed light on the hydrological changes associated with pre- and post-tree thinning treatments.”

L91: This section needs to be improved. Correctly indicate in detail how the different water fluxes were measured and sampled, as well as add how the analysis of the stable isotopes and SF6 were done and the accuracy of the measurements. Add the unit of each tracer. Also add the definition and calculation of the d-excess.

L109: described “by”

L109: What atmospheric information? Where does this come from?

L115-117: It is not clear to me how this was done “Using the average SF6 concentration of the air in the study area during the observation 115 period, the northern hemisphere data were proportionally distributed and corrected, and 116 a correction line (Mt. Karasawa line) was generated” From where we obtained the values to have an average SF6 concentration of the air?

L176: This sentence is incomplete?

L177: Does this mean that you also took air samples in the study area? See L52-53.

L217 (Figure 3): You could indicate the moment when forest thinning occurred, perhaps with a gray area.

L221: Change to “were large ranging from -1‰ to 23‰”

L235 and L244: I wonder if "water loss" is the correct term, I agree that a part is lost by evapotranspiration, but another part is still stored in the subsoil.

L240-243: I do not know to what extent these changes in discharge can be attributed with certainty to the effect of thinning. Whether reducing forest cover increases or decreases flow is still a matter of scientific debate. Two “recent” global analyses conclude that deforestation increases annual water yield (see https://doi.org/10.1002/eco.1838 , https://doi.org/10.1016/j.jhydrol.2016.12.040) however, there are studies that have found negligible or opposite effects (see https://doi.org/10.1111/j.1365-2486.2011.02589.x). In addition, it is evident that those effects are dependent on the characteristics of each watershed (e.g., https://doi.org/10.1002/hyp.10107)

L264-267 and L270-271: This text is unnecessary, as it repeats what is shown in Figure 4.

L291: This study is interesting, but an important issue is that the authors use the isotopic signal of rainfall, whereas in forested watersheds, rainfall interception by tree canopies redistribute rainwater as throughfall and stemflow, which are isotopically distinct from rainfall (please refer to https://doi.org/10.1002/hyp.14641 ) I am sure that using the isotopic signal of throughfall (which is the major water input) will lead to either different or relatively close results.

Author Response

(The authors gave the same response as above.)

Round 2

Reviewer 2 Report

I want to thank the authors for responding to all my comments. I am fully satisfied with their responses.

Two minor comments:

1) How can you explain the fact that after about 10 years “long after thinning” the hydrological processes return to "before thinning" conditions. I think this is worth discussing in the manuscript.

2) Page 10, Lines 345-352: I suggest rewriting this paragraph as: "Although previous studies have shown that water transmitted by the canopy as throughfall or stemflow can have different isotopic composition than rainfall [23], throughfall isotopic shift is negligible in our study area (data not shown). MTTs of groundwater and streamflow have been effectively estimated by others using rainfall isotopic signal [10]. In addition, our rainfall data set, longer than that of throughfall, was more appropriate for estimating short- and long-term MTT.” -----Please revise it so that there are no grammatical or syntax errors.

Author Response

Thank you for giving us another the opportunity to revise and submit our manuscript “Long-term variations in spring water mean transit time in a forested headwater catchment in Japan” for publication for the Water journal. Please accept our sincere thanks for your comments on the revised manuscript. Our point-by-point responses to the reviewers’ comments are included in the attachment.
